# Multilevel correlates of abdominal obesity in adolescents and youth living with HIV in peri-urban Cape Town, South Africa

**Monika Kamkuemah**[1ø¤]*, **Blessings Gausi**[1], **Tolu Oni**[1,2ø], **Keren Middelkoop**[3ø]

1 Research Initiative for Cities Health and Equity (RICHE), Division of Public Health Medicine, School of Public Health and Family Medicine, University of Cape Town, Cape Town, South Africa, 2 Medical Research Council Epidemiology Unit, University of Cambridge, Cambridge, United Kingdom, 3 Desmond Tutu HIV Centre, Institute of Infectious Disease & Molecular Medicine, Department of Medicine, Faculty of Health Sciences, University of Cape Town, Cape Town, South Africa

ø These authors contributed equally to this work.
¤ Current address: Innovation Africa and Department of Architecture, Faculty of Engineering, Built Environment and Information Technology, University of Pretoria, Pretoria, South Africa
* monika.kamkuemah@gmail.com

## Abstract

### Background

Chronic non-communicable disease comorbidities are a major problem faced by people living with HIV (PLHIV). Obesity is an important factor contributing to such comorbidities and PLHIV face an elevated risk of obesity. However, there is data paucity on the intersection of obesity and HIV in adolescents and youth living with HIV (AYLHIV) in sub-Saharan Africa. We therefore aimed to investigate the prevalence of abdominal obesity and associated multilevel factors in AYLHIV in peri-urban Cape Town, South Africa.

### Methods

We conducted a cross-sectional study enrolling AYLHIV aged 15–24 years attending primary healthcare facilities in peri-urban Cape Town in 2019. All measures, except for physical examination measures, were obtained via self-report using a self-administered electronic form. Our outcome of interest was abdominal obesity (waist-to-height ratio $\geq$ 0.5). We collected individual-level data and data on community, built and food environment factors. Data was summarized using descriptive statistics, stratified by obesity status. Multilevel logistic regression was conducted to investigate factors associated with abdominal obesity, adjusted for sex and age.

### Findings

A total of 87 participants were interviewed, 76% were female and the median age was 20.7 (IQR 18.9–23.0) years. More than two fifths had abdominal obesity (41%; 95% CI: 31.0–51.7%), compared to published rates for young people in the general population (13.7–22.1%). In multilevel models, skipping breakfast (aOR = 5.42; 95% CI: 1.32–22.25) was associated with higher odds of abdominal obesity, while daily wholegrain consumption

**Data Availability Statement:** All data files are available from the figshare database (https://doi.org/10.6084/m9.figshare.19204929.v1).

**Funding:** This work was supported by a Bristol-Myers Squibb Foundation grant awarded to TO (grant number 430960). https://www.bms.com/about-us/responsibility/bristol-myers-squibb-foundation/our-key-initiatives/secure-the-future.html TO is also funded by the National Institute for Health Research (NIHR) (16/137/34) using UK aid from the UK Government to support global health research. https://www.nihr.ac.uk/ MK was supported by the South African National Research Foundation (NRF: https://www.nrf.ac.za/) under joint funding with the German Academic Exchange Service (DAAD: https://www.daad.de/en/) for her PhD. The funders had no role in study design, data collection and analysis, decision to publish, or preparation of the manuscript.

**Competing interests:** The authors have declared that no competing interests exist.

(aOR = 0.20; 95% CI: 0.05–0.71) and weekly physical activity (aOR = 0.24; 95% CI: 0.06–0.92) were associated with lower odds of abdominal obesity. Higher anticipated stigma was associated with reduced odds of obesity (aOR = 0.58; 95% CI: 0.33–1.00). Land-use mix diversity (aOR = 0.52; 95% CI: 0.27–0.97), access to recreational places (aOR = 0.37; 95% CI: 0.18–0.74), higher perceived pedestrian and traffic safety (aOR = 0.20; 95% CI: 0.05–0.80) and having a non-fast-food restaurant within walking distance (aOR = 0.30; 95% CI: 0.10–0.93) were associated with reduced odds of abdominal obesity. The main limitations of the study were low statistical power and possible reporting bias from self-report measures.

## Conclusions

Our findings demonstrate a high prevalence of abdominal obesity and highlight multilevel correlates of obesity in AYLHIV in South Africa. An intersectoral approach to obesity prevention, intervening at multiple levels is necessary to intervene at this critical life stage.

## Introduction

Obesity is a major global health challenge and the leading risk factor for chronic non-communicable diseases (NCDs) including hypertension, cardiovascular disease, diabetes, several cancers and osteoporosis [1]. Obesity rates are increasing globally, particularly in sub-Saharan Africa (SSA) where the prevalence of overweight increased from 6% in 1990 to 21% in 2015 [2]. South Africa has the highest prevalence of overweight and obesity in SSA, with up to 70% of women and 33% of men classified as overweight or obese [3]. As with adults, South Africa has the highest prevalence of childhood overweight and obesity in Africa with 19% of boys and 26% of girls under 20 years classified as overweight or obese, rivalling that of many high-income countries [3].

An important factor in the prevalence of obesity is urbanisation: increased urbanisation is associated with lower levels of work-related physical activity, decreased levels of active transport, decreased energy expenditure during leisure time and increased consumption of refined and processed foods [4, 5]. Furthermore, environmental attributes like neighbourhood walkability, access to recreational spaces and pedestrian infrastructure affect willingness and ability to safely walk and engage in physical activity [6]. Rapid urbanisation and population growth in cities may result in increased crime rates, low air quality and destruction of recreational areas and green spaces, reducing walkability and opportunities to engage in physical activity [7]. In South Africa, 66% of the population were living in urban areas in 2018 [8], with national statistics showing that young people are especially mobile [9]. The urban built and food environments in low and middle-income countries (LMICs), where an increasing number of young people live, are increasingly obesogenic, promoting high energy intake and sedentary behaviour [10, 11]. Urban food environments, with supermarkets, food vendors, fast-food outlets and restaurants, facilitate access to a variety of foods, but micronutrient poor, energy-dense foods, which tend to be cheaper, are usually in over supply in urban areas [12]. For the urban poor, especially young people, the most easily available and affordable diets are often comprised of unhealthy, calorie-dense foods [10].

In addition to a growing obesity epidemic, South Africa has the largest antiretroviral therapy (ART) program in the world and the highest reported burden of adolescent HIV globally [13, 14]. Although wasting and thinness were visible markers of HIV-infection before the

advent of highly active ART, obesity has now been described as the latest epidemic in people living with HIV (PLHIV) who face an elevated risk of obesity resulting from a combination of psychosocial factors and the complications of long-term ART [15, 16]–risks that have also been identified in AYLHIV [17, 18]. However, environmental factors like urbanisation and the built and food environments, important drivers of obesity in LMICs, are understudied in this subgroup of AYLHIV. We therefore set out to investigate the prevalence of abdominal obesity and associated individual, household, community and neighbourhood environmental factors in AYLHIV in peri-urban Cape Town, South Africa.

## Methods

### Study design and setting

We conducted a cross-sectional study enrolling AYLHIV aged 15–24 years attending primary healthcare facilities in peri-urban Cape Town between March and December 2019. Cape Town, based in the Western Cape province, is the second biggest metropolitan city in South Africa [19] and in 2016, adolescents and youth aged 15–24 years comprised 16.3% of the >7 million people living in the province [20]. The primary healthcare facilities selected serve a catchment population living in peri-urban, high-density, low-income townships, collectively known as the Cape Flats [21]. Health facilities in the City of Cape Town fall within eight health sub-districts, namely Eastern, Western, Northern, Southern, Khayelitsha, Klipfontein, Tygerberg and Mitchells Plain [22]. Sampling and recruitment were conducted via convenience sampling during routine clinic visits, with the aim of recruiting across all eight of the City of Cape Town's health districts. Procedures have been previously described in detail and recruitment by sub-district and facility is included as a supplementary figure (S1 Fig) [23].

Ethical clearance was obtained from the Human Research Ethics Committee at the University of Cape Town (HREC ref no: 520/2017) and approval to access the facilities was obtained from Provincial and Local Government Departments of Health. Written informed consent (or assent with parental/ caregiver consent for participants less than 18 years old) was obtained for all participants.

### Measures

All measures, except for physical examination measures, were obtained via self-report using a self-administered electronic form on a hand-held Android device. Physical examinations were conducted by trained research staff. (*A detailed table of measurements and variable descriptions is included as a supplementary table in S1 Table*).

**Individual-level variables.** *Socio-demographics*. The following socio-demographic characteristics were collected: age (in years), sex, socio-economic status, educational attainment and absence from school, history of pregnancy/ impregnating someone and number of children. Age was categorised as follows: 15–17, 18–19, 20–21 and 22–24 years. Socio-economic status was measured using the Youth Multidimensional Poverty Index (YMPI) which consists of eleven indicators across five dimensions: general health and functioning status, educational attainment, living standards, asset deprivation and economic opportunities [24]. A deprivation score was calculated for each dimension and an overall composite score was derived from the weighted indicators. An individual is identified as being multidimensionally poor–MPI poor– if they are deprived in a third or more of the weighted indicators, with a composite score $\geq$ 33.3% [25].

*Clinical characteristics*. The following clinical characteristics were collected: anthropometrics [height (in cm), weight (in kg), waist circumference (in cm)], blood pressure (in mmHg) and family history of chronic conditions. Sitting blood pressure (BP) was measured

using a ROSSMAX automatic blood pressure monitor (*Rossmax (Shanghai) Incorporation Ltd*). Two readings were taken at least two minutes apart and the average was computed. Elevated blood pressure and hypertension were classified according to the South African Hypertension Practice guidelines as follows: normal BP: systolic BP (SBP) < 130 mmHg and diastolic BP (DBP) < 85 mmHg; elevated BP: SBP 130–139 mmHg / DBP 85–89 mmHg; hypertension: SBP 140–159 mmHg / DBP 90–99 mmHg [26]. Family history of chronic conditions was assessed via self-report and included conditions such as diabetes, stroke and hypertension.

Our main outcome of interest was abdominal obesity status (waist-to-height ratio $\geq 0.5$ [27]). While BMI is the most widely used adult, population-level measure of overweight and obesity, it may not correspond with body fat percentage in different populations like PLHIV who are subject to visceral adiposity [27]. Measures of abdominal obesity may be more sensitive in detecting changes caused by changes in medication and immunosuppression, compared to BMI and hence better at detecting PLHIV who are at increased cardiometabolic risk [28]. Furthermore, waist circumference and waist-to-height ratio are better predictors of cardiovascular disease risk in children and adolescents than BMI [29]. Height and weight were measured using a sliding balance weight-and-height measuring scale with participants barefoot and wearing light clothing. Height was measured to the nearest 0.5 cm and weight to the nearest 0.1 kg. Waist circumference was measured using stretch-resistant measuring tape according to the WHO STEPS Protocol [27]. Readings were taken to the nearest 0.1 cm. Two measurements were taken from which an average was computed for analysis. For weight, height, and waist circumference, if the two readings differed by more than 100g, 2cm and 0.1cm respectively, a third measurement was taken, and the two closest measurements were recorded and an average of these computed.

*Knowledge and behaviour*. Dietary intake was assessed using a 23-item food frequency questionnaire (FFQ) adapted from the Health Behaviour in School-aged Children Survey [30]. For this analysis, we reported on the median weekly portions consumed and daily consumption of fruit, vegetables, wholegrains, fast-foods, deep-fried foods, cakes and biscuits and sugar-sweetened beverages (SSBs). Other dietary behaviours assessed were skipping breakfast, consuming meals prepared outside the home and school lunch consumption for those currently in school. Nutritional knowledge was assessed using the revised *General Nutrition Knowledge Questionnaire* (GNKQ-R) [31]. The questionnaire consists of 88 items divided into four sections: dietary recommendation (18 items), food groups (36 items), healthy food choices (13 items) and diet, disease, and weight associations (21 items).

Physical activity (PA) was assessed using *the International Physical Activity Questionnaire* (IPAQ) short form using the last seven days self-administered format [32]. PA was further dichotomised into insufficient PA (< 600 Metabolic Equivalents of Task (MET) minutes/ week) or sufficient PA. Sedentary behavior was dichotomised as present or absent, the former defined as spending three or more hours per day watching television, playing computer games or other sitting activities according to the Global School-based Student Health Survey criteria [33]. We also assessed whether active transport (walking / cycling) was part of participants' daily commute.

**Household-level variables.** We collected information on physical dwelling characteristics, thermal comfort in the home, food security, orphanhood status and family structure. Dwelling characteristics were assessed according to the *2011 South African census questionnaire* [34]: housing informality, access to amenities, sanitation, primary source of water, household waste removal and history of flooding / fire or other adverse events. Thermal comfort was assessed using self-report measures of perceived thermal comfort, asking whether the participant experienced any seasonal discomfort in the home on a scale ranging from never, rarely, sometimes,

often or permanently [35]. Food security was measured using the *Household Food Insecurity Access Scale* (HFIAS) which provides a continuous measure of the degree of food insecurity experienced in a household in the previous month [36]. The scores were tallied, and the level of food security was categorised into mild, moderate and severe food insecurity according to the HFIAS scoring protocol [36].

**Community-level variables.** We collected information on experiences of stigma, neighbourhood social capital, crime safety and exposure to violence in the community. Stigma was measured using the *HIV Stigma Scale for Adolescents Living with HIV* (ALHIV-SS) [37]. This scale includes elements pertaining to anticipated, internalised, and enacted stigma. Neighbourhood social capital was measured using neighbourhood trust, friendliness, belonging and reciprocity. We created a dichotomous variable for each social capital response [38]. Crime safety items from the *Neighbourhood Environment Walkability Scale for Youth* (NEWS-Y) scale [39] were used to measure perceptions of neighbourhood crime. A mean composite score was computed from the Crime Safety subscale of the NEWS-Y giving an overall score. Exposure to violence was measured using eight sub-items from the *Survey of Exposure to Community Violence scale* [40]. Violence was categorised as no violence (score < 2), moderate level (score 2–3), and high level of violence (score 4–8).

**Neighbourhood environment-level variables.** The neighbourhood built and food environments were assessed using the NEWS-Y scale which measures participants' perceptions of walkability in their immediate neighbourhood [39]. The scale has nine subscales each with a set of indicators and response options which were summarised using Z-scores. An overall walkability score was created by calculating and summing Z-scores for each of the nine subscales. Higher scores indicate a more walkable environment. Accessibility questions from the land-use mix diversity subscale were used to assess the food environment, with walking distance defined as stores or facilities within a 20-minute walk or less from home [39].

## Statistical analysis

Data were summarised using descriptive statistics, stratified by obesity status. Differences between variables were compared by abdominal obesity status using Pearson's $\chi 2$ goodness of fit tests and Fisher's exact tests for categorical variables. We used visual displays, histograms, box plots and the Shapiro–Wilk test to test for normality of the continuous variables. Continuous, non-parametric variables were compared using the Wilcoxon rank-sum test while normally distributed variables were compared using t-tests. All statistical analyses were done using Stata (version 14) (*Stata Corporation*, *College Station*, *Texas*, *USA*). We explored relationships between individual-level, household-level, community-level and environment-level variables and abdominal obesity using crude odds ratios (ORs) derived from bivariate logistic regression models and adjusted ORs (aOR) from multilevel logistic models adjusted for age and sex. Variables found to be associated with abdominal obesity in bivariate analysis (p < 0.10) and variables identified a priori in the literature were included in the multilevel models. The following variables identified a priori to be associated with obesity were included in the multilevel models: multidimensional poverty (YMPI), blood pressure, family history of diabetes, sedentary behaviour, physical activity, dietary intake, household food security, and level of community violence. The multilevel data structure consisted of participants (level 1) nested within sub-districts which were used as a proxy for neighbourhoods (level 2). We checked the quality-of-fit for all models using the likelihood-ratio (LR) test and tested the underlying model assumptions of linearity, homoscedasticity, and normal distribution of the residuals. We also checked the intra-class correlation coefficient (ICC) to analyse the variability within and between sub-districts. Statistical significance was set at p < 0.05 for the

multilevel analyses. Crude and adjusted odds ratios are presented with confidence intervals and p-values.

## Results

### Individual and household characteristics

**Sociodemographic characteristics.** A total of 87 participants were interviewed, with median age 20.7 years (IQR 18.9–23.0) and 76% were female. Overall, 27% were not in education, employment, or training (NEET) at the time of the study and 43% were multidimensionally poor.

Forty one percent of participants met the primary outcome criteria for abdominal obesity (95% CI 31.0–51.7%). According to BMI status, 24% of participants were overweight and 11% had obesity. Notably, 24% of those with normal BMI had abdominal obesity. Almost a fifth (18%) of participants had elevated blood pressure and 6% had hypertension, while 28% self-reported a family history of diabetes. Table 1 displays a summary of individual, household and clinical characteristics stratified by abdominal obesity status.

### Physical activity, dietary behaviour, and nutrition knowledge

The majority reported engaging in some form of physical activity and 72% used active transport as part of their daily commute (Table 2). Two-thirds met the criteria for sufficient levels of physical activity per week of > 600 MET-minutes per week, while half spent three or more hours sedentary in a typical day. More than a quarter of participants (26%) reported daily consumption of fruits, 51% reported daily vegetable consumption and 41% ate wholegrains daily. More than a quarter (28%) reported consuming deep-fried foods, 27% drank SSBs, and 34% ate sweets and cakes daily, while 20% consumed fast-foods daily or more than once daily. Forty percent of participants skipped breakfast frequently or almost every day in the week. Participants scored an average of 37.7% on the GNKQ-R (95% Confidence Interval (CI): 35.5–39.9%). Other dietary and behavioural characteristics are reported in Table 2.

### Community characteristics

Overall, 43% of participants reported belonging to an extra-mural group in their community (Table 3). The majority reported high levels of neighbourhood reciprocity (79%), friendliness (87%) and belonging (77%). Participants reported experiencing low levels of stigma. Overall, 61% of participants were exposed to high levels of violence. However, only a quarter of participants perceived their neighbourhoods as risky or unsafe to walk in at night.

### Neighbourhood environment characteristics

The average neighbourhood walkability scores are reported in Table 3. The highest scoring domain was residential density: mean 3.71 (SD, 0.71), followed by crime safety and land-use mix diversity. The lowest scoring domains were neighbourhood aesthetics and street connectivity. Over 80% had access to a small grocer or fruit and vegetable market within walking distance from home, while supermarkets and non-fast-food restaurants were less accessible (59% and 50% respectively).

Fig 1 is an illustrative display of the direction of effect of the multilevel factors found to be associated with abdominal obesity in bivariate and multivariate analysis. These factors are discussed in more detail below.

**Table 1. Sociodemographic and clinical characteristics of AYLHIV by abdominal obesity status.**

| Variable: median (IQR) or n (%) | | Non-obese | Obese | Total n = 87 |
|---|---|---|---|---|
| | | n = 51 (59%) | n = 36 (41%) | |
| **Sociodemographic characteristics** | | | | |
| Age (years) | | 20.2 (18.6–22.7) | 21.7 (19.6–23.5) | 20.7 (18.9–23.0) |
| Age distribution | 15–17 years | 11 (22%) | 2 (6%) | 13 (15%) |
| | 18–19 years | 12 (24%) | 9 (25%) | 21 (24%) |
| | 20–21 years | 13 (25%) | 11 (31%) | 24 (28%) |
| | 22–24 years | 15 (29%) | 14 (39%) | 29 (33%) |
| Gender* | Male | 16 (76%) | 5 (24%) | 21 (24%) |
| | Female | 35 (53%) | 31 (47%) | 66 (76%) |
| Ever pregnant/ impregnated someone* | | 7 (14%) | 13 (37%) | 20 (23%) |
| Number of children* | 0 children | 45 (88%) | 27 (75%) | 72 (83%) |
| | 1 child | 6 (12%) | 8 (22%) | 14 (16%) |
| | 2 children | 0 (0%) | 1 (3%) | 1 (1%) |
| Educational attainment | Primary school | 2 (4%) | 1 (3%) | 3 (3%) |
| | Some secondary school | 25 (49%) | 19 (53%) | 44 (51%) |
| | Completed matric/ equivalent | 19 (37%) | 12 (33%) | 31 (36%) |
| | Some tertiary education | 5 (10%) | 4 (11%) | 9 (10%) |
| Ever repeated a grade at school | | 24 (47%) | 21 (58%) | 45 (52%) |
| Days absent from school or work in past month (n = 86) | 0 days | 31 (62%) | 14 (39%) | 45 (52%) |
| | 1–2 days | 17 (34%) | 13 (36%) | 30 (35%) |
| | 3 or more days | 2 (4%) | 9 (24%) | 11 (12%) |
| Prevalence of multidimensional poverty | | 18 (37%) | 18 (53%) | 36 (43%) |
| **Clinical characteristics** | | | | |
| Waist circumference (WC) in cm* | | 72.25 (68–75) | 89 (82–102) | 76 (71–87) |
| BMI in kg/m$^2$ * | | 20.25 (18.93–22.22) | 27.12 (24.45–31.31) | 22.55 (19.59–26.45) |
| **Blood Pressure category** | Normal BP: SBP<130 & DBP<85 | 41 (80%) | 25 (69%) | 66 (76%) |
| | Elevated BP: SBP 130–139/ DBP 85–89 | 7 (14%) | 9 (25%) | 16 (18%) |
| | HPT: SBP 140–159/ DBP 90–99 | 3 (6%) | 2 (6%) | 5 (6%) |
| **Family history of diabetes** | | 16 (31%) | 8 (22%) | 24 (28%) |
| **Housing characteristics** | | | | |
| Dwelling type | Formal dwelling | 33 (66%) | 24 (67%) | 57 (66%) |
| (n = 86) | Informal dwelling | 17 (34%) | 12 (33%) | 29 (34%) |
| Time lived in current residence in years (n = 81) | | 9.5 (4–19) | 8 (2–18) | 9 (3–18) |
| Residential stability in lifetime | never moved | 32 (63%) | 21 (58%) | 53 (61%) |
| | moved once | 14 (27%) | 8 (22%) | 22 (25%) |
| | moved twice | 3 (6%) | 5 (14%) | 8 (9%) |
| | moved three or more times | 2 (4%) | 2 (6%) | 4 (5%) |
| Dwelling ever damaged by flooding, fire, or another negative event (n = 81) | | 8 (17%) | 7 (20%) | 15 (19%) |
| Assets deprivation$^\alpha$ | | 12 (24%) | 10 (28%) | 22 (25%) |
| **Access to amenities and food security** | | | | |
| Using paraffin, candles, nothing / other for lighting | | 0 (0%) | 1 (3%) | 1 (1%) |
| Using paraffin, wood, coal, nothing / other for heating | | 15 (29%) | 10 (28%) | 25 (29%) |
| Using paraffin, wood, coal, dung / other for cooking | | 0 (0%) | 1 (3%) | 1 (1%) |
| Households without access to flush toilet | | 4 (8%) | 3 (8%) | 7 (8%) |
| Source of water | Households without piped water on site | 9 (18%) | 5 (14%) | 14 (16%) |
| | Piped water inside dwelling | 33 (65%) | 21 (58%) | 54 (62%) |
| | Piped water on site or in yard | 9 (18%) | 10 (28%) | 19 (22%) |

(*Continued*)

**Table 1.** (Continued)

| Variable: median (IQR) or n (%) | | Non-obese | Obese | Total n = 87 |
|---|---|---|---|---|
| | | n = 51 (59%) | n = 36 (41%) | |
| Household waste removal | Removed weekly by local authorities | 30 (67%) | 25 (71%) | 55 (69%) |
| (n = 80) | Removed less often than once a week | 5 (11%) | 1 (3%) | 6 (8%) |
| | Communal refuse dump | 3 (7%) | 3 (9%) | 6 (8%) |
| | Own refuse dump | 7 (15%) | 6 (17%) | 13 (16%) |
| Thermal discomfort [β] | Summer | 16 (35%) | 17 (49%) | 33 (41%) |
| (n = 81) | Autumn or spring | 13 (28%) | 19 (54%) | 32 (40%) |
| | Winter | 25 (54%) | 25 (71%) | 50 (62%) |
| Household food security | Food secure | 15 (29%) | 12 (33%) | 27 (31%) |
| | Mild food insecurity | 5 (10%) | 5 (14%) | 10 (11%) |
| | Moderate food insecurity | 12 (24%) | 6 (17%) | 18 (21%) |
| | Severe food insecurity | 19 (37%) | 13 (36%) | 32 (37%) |
| **Family structure** | | | | |
| Orphanhood status | Both parents alive | 18 (35%) | 13 (36%) | 31 (36%) |
| | Death of a parent | 29 (57%) | 22 (61%) | 51 (59%) |
| | Don't Know | 4 (8%) | 1 (3%) | 5 (6%) |
| Deceased parent | Mother | 11 (38%) | 5 (24%) | 16 (32%) |
| (n = 50) | Father | 12 (41%) | 12 (57%) | 24 (48%) |
| | Both parents deceased | 6 (21%) | 4 (19%) | 10 (20%) |
| Primary caregiver | Biological parent(s) | 22 (43%) | 18 (50%) | 40 (46%) |
| | Legal guardian/ adoptive parent | 3 (6%) | 1 (3%) | 4 (5%) |
| | Grandparent(s) | 7 (14%) | 1 (3%) | 8 (9%) |
| | Relative (aunt, uncle, etc.) | 9 (18%) | 9 (25%) | 18 (21%) |
| | Other | 10 (20%) | 7 (19%) | 17 (20%) |
| Number of people residing in same house | | 5 (4–6) | 5 (4–6) | 5 (4–6) |
| **Number of working-age adults employed in the household** (aged 18–64 years) | | 2 (1–3) | 2 (1–2.5) | 2 (1–3) |

[¥] Fishers exact test p-value

[α] Individual living in a household that does not own more than two of: radio, television, landline, cell phone, bike, motorbike, or refrigerator AND does not own a motor car or truck

[β] Thermal discomfort experienced in the home sometimes, often or permanently.

## Individual-level factors

Bivariate models showed an age gradient by abdominal obesity status with the odds of abdominal obesity increasing with age as shown in Table 4. Females had four-fold increased odds of abdominal obesity compared to males. The multilevel model including age and sex, was not significantly different from the null model ($p = 0.124$) but given that age and sex are clinically significant confounders, we included them in subsequent models. The model with age and sex is hereafter referred to as model 2. Other clinical characteristics, blood pressure and family history of diabetes were not significantly associated with abdominal obesity in bivariate or multilevel analysis. The ICC ranged from 0.01 to 0.2, indicating some level of variability between and within sub-districts.

Participants who engaged in at least ten minutes of moderate-intensity physical activity per week had 76% reduced odds of abdominal obesity compared to those who did not engage in physical activity (aOR = 0.24; 95% CI: 0.06–0.92). Including moderate-intensity physical activity improved on model 2 ($p = 0.028$). Those who skipped breakfast had higher odds of abdominal obesity compared to those who ate breakfast on five or more days per week, an association

**Table 2. Physical activity, dietary behaviour, and nutrition knowledge of AYLHIV by abdominal obesity status.**

| Variable: median (IQR) or n (%) | | Non-obese n = 46 (57%) | Obese n = 35 (43%) | Total n = 81 |
|---|---|---|---|---|
| **Physical activity** | | | | |
| Vigorous-intensity physical activity | Prevalence of PA for ≥10 minutes | 21 (46%) | 13 (37%) | 34 (42%) |
| | Time spent per day in minutes | 60 (30–150) | 105 (30–120) | 60 (30–120) |
| | MET minutes /week | 1560 (720–3840) | 2040 (480–4800) | 1800 (600–4320) |
| Moderate-intensity physical activity | Prevalence of PA for ≥10 minutes | 39 (85%) | 25 (71%) | 64 (79%) |
| | Time spent per day in minutes | 60 (30–120) | 35 (30–60) | 60 (30–90) |
| | MET minutes /week | 720 (480–1440) | 720 (520–1200) | 720 (480–1440) |
| Active transport [t] | Walking or cycling for ≥ 10 minutes | 34 (74%) | 24 (69%) | 58 (72%) |
| | Time spent walking/ cycling daily in minutes | 30 (30–60) | 50 (30–60) | 30 (30–60) |
| | Walking MET-minutes/week | 495 (280.5–1188) | 693 (396–1188) | 528 (297–1188) |
| Total physical activity MET-minutes/week | | 1314 (480–3396) | 1173 (560–3348) | 1207.5 (495–3372) |
| Insufficient physical activity (< 600 MET-minutes per week) | | 16 (36%) | 10 (29%) | 26 (33%) |
| High physical activity (≥ 3000 MET minutes per week) | | 12 (27%) | 9 (26%) | 21 (26%) |
| Sedentary behaviour: 3 or more hours per day | | 21 (46%) | 19 (54%) | 40 (49%) |
| Currently enrolled in educational institution or working | | 31 (61%) | 21 (58%) | 52 (60%) |
| Travel time from home to school or work (n = 52) | 1–5 min | 4 (13%) | 1 (5%) | 5 (10%) |
| | 6–10 min | 1 (3%) | 2 (10%) | 3 (6%) |
| | 11–20 min | 8 (26%) | 3 (14%) | 11 (21%) |
| | 21–30 min | 10 (32%) | 7 (33%) | 17 (33%) |
| | 31+ min | 8 (26%) | 8 (38%) | 16 (31%) |
| Additional transport mode (more than one means) | | 12 (40%) | 10 (48%) | 22 (42%) |
| **Food Frequency** | | | | |
| **Weekly portions of fruit consumed: median (IQR)** | | **3 (0.5–7)** | **3 (1–3)** | **3 (1–7)** |
| n = 77 | Daily or more than once daily | 15 (34%) | 5 (15%) | 20 (26%) |
| **Weekly portions of vegetables consumed: median (IQR)** | | **7 (3–7)** | **7 (3–10)** | **7 (3–10)** |
| n = 76 | Daily or more than once daily | 22 (51%) | 17 (52%) | 39 (51%) |
| **Weekly portions of wholegrain consumed: median (IQR)** | | **7 (1–10)** | **2 (0.5–6.25)** | **5.5 (1–7)** |
| n = 64 | Daily or more than once daily | 19 (53%) | 7 (25%) | 26 (41%) |
| **Weekly portions of deep-fried foods consumed: median (IQR)** | | **3 (0.5–7)** | **1 (0.5–7)** | **2 (0.5–7)** |
| n = 64 | Daily or more than once daily | 10 (26%) | 8 (31%) | 18 (28%) |
| **Weekly portions of fast foods consumed: median (IQR)** | | **1 (0.5–5.5)** | **0.75 (0.5–4.25)** | **1 (0.5–5.5)** |
| n = 65 | Daily or more than once daily | 9 (24%) | 4 (14%) | 13 (20%) |
| **Weekly portions of SSBs consumed: median (IQR)** | | **3 (0.5–7)** | **3 (1–7)** | **3 (0.5–7)** |
| n = 74 | Daily or more than once daily | 11 (26%) | 9 (28%) | 20 (27%) |
| **Weekly portions of sweets and cakes consumed: median (IQR)** | | **3 (0.5–7)** | **3 (1–10)** | **3 (1–7)** |
| n = 76 | Daily or more than once daily | 14 (33%) | 12 (36%) | 26 (34%) |
| **Dietary Behaviour and Knowledge** | | | | |
| Ate meal prepared outside the home in past week (n = 74) | | 28 (68%) | 23 (70%) | 51 (69%) |
| Meals eaten outside home in past week: median (IQR) | | 2 (1.5–4.5) | 2 (2–3) | 2 (2–4) |
| Breakfast consumption (n = 77) ** | Skippers: breakfast 0–2 days/week | 7 (16%) | 10 (30%) | 17 (22%) |
| | Semi-skippers: breakfast 3–4 days/week | 4 (9%) | 10 (30%) | 14 (18%) |
| | Non-skippers: breakfast 5–7 days/week | 33 (75%) | 13 (39%) | 46 (60%) |
| School lunch (n = 38) [π] | School serves lunch ** | 16 (73%) | 6 (38%) | 22 (58%) |
| School lunch in the week (n = 22) | Never | 4 (25%) | 0 | 4 (18%) |
| | Occasionally (1–3 days) | 6 (37.5%) | 3 (50%) | 9 (41%) |
| | Always (4–5 days) | 6 (37.5%) | 3 (50%) | 9 (41%) |
| **General Nutrition Knowledge (% score/ 88) mean (95% CI)** | | 37.8 (34.9–40.7) | 37.7 (34.1–41.3) | 37.7 (35.5–39.9) |

*(Continued)*

**Table 2.** (Continued)

| Variable: median (IQR) or n (%) | | Non-obese n = 46 (57%) | Obese n = 35 (43%) | Total n = 81 |
|---|---|---|---|---|
| n = 76 | 1. Dietary recommendations (% score/18) | 44.3 (39.4–49.2) | 42.4 (36.9–47.8) | 43.5 (39.9–47.1) |
| | 2. Food Groups (% score/36) | 38.2 (35.1–41.3) | 37.8 (33.7 41.9) | 38.0 (35.6–40.4) |
| | 3. Healthy Food choices (% score/13) | 31.8 (26.7–36.9) | 33.7 (26.5–40.8) | 32.6 (28.5–36.7) |
| | 4. Diet, disease relationships (% score/21) | 34.3 (30.0–38.6) | 36.0 (31.1–40.9) | 35.0 (31.9–38.2) |

PA = physical activity, MET = metabolic equivalent of task, SSB = sugar sweetened beverages.

*p-value from Wilcoxon rank-sum (Mann-Whitney) test

**p-value from Fisher's exact test

[t] Only reported walking, none used bicycles or motorcycles for active transport.

[π] in primary or high school part of National School Nutrition Program.

which emerged in the multilevel model as well (aOR = 5.42; 95% CI: 1.32–22.25). Those who ate fruit and wholegrains daily had lower odds of abdominal obesity compared to those who ate these less frequently. However, daily fruit intake was not significantly associated with abdominal obesity in the adjusted multilevel model. The model with daily wholegrain intake improved on model 2 (p = 0.006) but was imprecise. Participants attending a school serving school lunch had lower odds of obesity in bivariate analysis, (OR = 0.23, 95% CI: 0.057–0.89), but this association was not statistically significant in the adjusted multilevel analysis. While those who were absent from school or work on one or more days in the past month had increased odds of abdominal obesity compared to those who were not absent from school or work (OR = 2.56; 95% CI: 1.06–6.18). This association also emerged in the multilevel model after adjusting for age and sex and the covariance structure (aOR = 3.06; 95% CI: 1.11–8.40).

## Household-level factors

Household food security and orphanhood status were not significantly associated with abdominal obesity in bivariate analysis. Experiencing thermal discomfort in the home in autumn or spring was associated with four-fold increased odds of abdominal obesity (aOR = 4.42; 95% CI: 1.43–13.73). The model with thermal discomfort experienced in the home improved on model 2. Other measures of the home environment, multidimensional poverty and access to amenities were not significantly associated with abdominal obesity in bivariate or multilevel analysis as shown in Table 5.

## Community-level factors

Measures of neighbourhood social capital were not significantly associated with abdominal obesity, apart from neighbourhood belonging which showed a tendency towards increased risk of abdominal obesity in those who reported higher neighbourhood belonging (aOR = 2.68; 95% CI: 0.81–8.89), although these odds ratios were not statistically significant (see Table 5). Those with higher anticipated stigma had 42% reduced odds of having abdominal obesity compared to those with lower anticipated stigma from the community (aOR = 0.58; 95% CI: 0.33–1.00).

## Perceived built and food environment factors

Participants with perceived higher land-use mix diversity had 48% reduced odds of having abdominal obesity compared to those with lower neighbourhood diversity (aOR = 0.52; 95% CI: 0.27–0.97). Those with perceived better access to recreational spaces had 63% reduced

**Table 3. Community and built environment characteristics of AYLHIV by abdominal obesity status.**

| Variable: Median (IQR) or n (%) | | Non-obese | Obese | Total |
|---|---|---|---|---|
| | | n = 51 | n = 36 | n = 87 |
| **Social networks:** Belong to extra-mural group (not run by school) | | 22 (43%) | 15 (42%) | 37 (43%) |
| **Social Capital** | | | | |
| | High neighbourhood trust | 35 (69%) | 21 (58%) | 56 (64%) |
| | High neighbourhood reciprocity | 41 (80%) | 28 (78%) | 69 (79%) |
| | High neighbourhood friendliness | 45 (88%) | 31 (86%) | 76 (87%) |
| | High neighbourhood belonging | 36 (71%) | 31 (86%) | 67 (77%) |
| **Experiences of stigma:** Anticipated (2 items), Enacted (3 items), Internalised (5 items) | | | | |
| | Anticipated stigma (max = 8) *(p = 0.042[fi])* | 2 (1.5–3) | 1.5 (1–2.5) | 2 (1–2.5) |
| | Enacted stigma (max = 12) | 1 (1–1.67) | 1 (1–1.67) | 1 (1–1.67) |
| | Internalised stigma (max = 20) | 1.6 (1.2–2) | 1.6 (1–2.4) | 1.6 (1–2) |
| **Exposure to Community Violence (maximum score = 8)** | | 4 (2–5) | 5 (3–5.5) | 4 (3–5) |
| Heard of any killing in your community | | 37 (73%) | 29 (81%) | 66 (76%) |
| Seen a dead body (not at a funeral) | | 23 (45%) | 18 (50%) | 41 (47%) |
| Afraid of anyone in your community or yard | | 18 (35%) | 17 (47%) | 35 (40%) |
| Seen someone pointing or shooting a gun at someone | | 21 (41%) | 15 (42%) | 36 (41%) |
| Seen someone other than police pointing or shooting a gun at someone | | 22 (43%) | 15 (42%) | 37 (43%) |
| Someone personally known to you has been: | Shot | 23 (45%) | 21 (58%) | 44 (51%) |
| | Stabbed | 29 (57%) | 24 (67%) | 53 (61%) |
| | Raped | 15 (29%) | 16 (44%) | 31 (36%) |
| **Exposure to violence category** | No or little violence (score 0–1) | 9 (18%) | 5 (14%) | 14 (16%) |
| | Moderate level of violence (score 2–3) | 14 (27%) | 6 (17%) | 20 (23%) |
| | High level of violence (score $\geq$ 4) | 28 (55%) | 25 (69%) | 53 (61%) |
| **Crime safety** (NEWS-Y subset [μ]) (n = 76) | Perceived high crime rate in neighbourhood | 10 (23%) | 9 (27%) | 19 (25%) |
| | Unsafe to walk at night | 9 (21%) | 10 (30%) | 19 (25%) |
| | Worried to be outside alone around home | 17 (40%) | 13 (39%) | 30 (39%) |
| | Worried to be outside with someone around home | 13 (30%) | 11 (33%) | 24 (31%) |
| | Worried to be or walk around neighbourhood alone or with friends | 16 (37%) | 13 (39%) | 29 (38%) |
| | Worried about being in a local or nearby park | 16 (37%) | 9 (27%) | 25 (33%) |
| **Built Environment: Neighbourhood Environment Walkability** | | | | |
| NEWS-Y Composite Score | A. Land use mix-diversity *(p = 0.0440)* | 2.97 (0.78) | 2.58 (0.92) | 2.80 (0.86) |
| | B. Access to recreational places *(p = 0.0032)* [Σ] | 2.78 (0.90) | 2.18 (0.76) | 2.53 (0.89) |
| Mean (± SD) | C. Residential density *(p = 0.0780)* | 3.84 (0.76) | 3.55 (0.62) | 3.71 (0.71) |
| | D. Land use mix-access (access to services) | 2.58 (0.63) | 2.51 (0.57) | 2.55 (0.60) |
| | E. Street connectivity | 2.31 (0.80) | 2.49 (0.72) | 2.39 (0.77) |
| | F. Walking or cycling facilities | 2.63 (0.88) | 2.55 (0.76) | 2.60 (0.83) |
| | G. Neighbourhood aesthetics *(p = 0.0406)* [Σ] | 2.46 (0.94) | 2.04 (0.77) | 2.27 (0.89) |
| | H. Pedestrian and traffic safety *(p = 0.0263)* [Σ] | 2.66 (0.43) | 2.45 (0.38) | 2.57 (0.42) |
| | I. Crime safety | 2.83 (0.89) | 2.94 (0.88) | 2.88 (0.88) |
| **Food Environment: Within walking distance from home to destination** [π] | | | | |
| Kiosk, corner store or small grocer (n = 73) | | 38 (86%) | 25 (86%) | 63 (86%) |
| Supermarket (n = 75) | | 29 (66%) | 15 (48%) | 44 (59%) |
| Fruit or vegetable market (n = 73) | | 38 (90%) | 27 (87%) | 65 (89%) |
| Fast food restaurant (n = 75) | | 27 (63%) | 17 (53%) | 44 (59%) |
| Non-fast-food restaurant (n = 74) *(p = 0.078)* | | 25 (50%) | 12 (39%) | 37 (50%) |

*(Continued)*

**Table 3.** (Continued)

| | | | |
|---|---|---|---|
| Coffee shop (n = 73) | 18 (43%) | 9 (30%) | 27 (37%) |

¥Fishers exact test p-value

Σ Kruskal-Wallis p-value

ᴴDichotomous variable for crime safety elements: strongly agreed or somewhat agreed versus somewhat disagreed or strongly disagreed

Φ Two-sample t-test with equal variances p-value

πWithin walking distance defined as ≤ 20-minute walk from home.

odds of having abdominal obesity compared to participants with lower perceived access to recreational spaces (aOR = 0.37; 95% CI: 0.18–0.74) (*p = 0.005*). Both models significantly improved on model 2 (*p = 0.029* and *p = 0.002 respectively*). Those with perceived higher pedestrian and traffic safety had 80% reduced odds of having abdominal obesity compared to those with lower perceived traffic safety (aOR = 0.20; 95% CI: 0.05–0.80) (*p = 0.023*). However, the model failed to converge, and we therefore could not compare it to model 2 using the LR test statistic. The models with residential density and neighbourhood aesthetics did not improve the model fit and their odds ratios were not significant in multilevel analysis. Other NEWS-Y items (street connectivity, places for walking and cycling, and crime safety) were not significant in bivariate analysis and were therefore not included in the multilevel models.

Participants with non-fast-food restaurants within walking distance had 70% reduced odds of having abdominal obesity compared to those without non-fast-food restaurants within walking distance (aOR = 0.30; 95% CI: 0.10–0.93). The model improved significantly on model 2 (*p = 0.027*). The model with "supermarket within walking distance from home" did not improve significantly on model 2 (*p = 0.078*) but suggests that those with supermarkets within walking distance had lower odds of abdominal obesity compared to those without access to a supermarket (aOR = 0.37; 95% CI: 0.12–1.16). The rest of the food environment variables did not improve on the model fit and were not statistically significant.

## Discussion

We investigated the prevalence and multilevel determinants of abdominal obesity in AYLHIV in peri-urban Cape Town. Overall, we found a 41% prevalence of abdominal obesity with numerous factors acting at multiple levels associated with abdominal obesity as displayed in **Fig 1**. Female AYLHIV and those who skipped breakfast had increased risk of abdominal obesity [41, 42], while weekly moderate-intensity physical activity and wholegrain consumption were protective from the risk of abdominal obesity as established in previous studies [43, 44]. Absence from school or work in the past month and experiencing thermal discomfort in the home emerged as unexpected factors associated with increased risk of abdominal obesity. At the community, built and food environment levels, anticipated stigma, land-use mix diversity, access to recreational places, pedestrian and traffic safety and having a non-fast-food restaurant within walking distance were associated with reduced odds of abdominal obesity in AYLHIV.

The finding of a strong association between female sex and obesity is in line with the literature on sex differences in obesity rates in LMICs [42, 45]. While we observed an age gradient with those in the older age groups more likely to have abdominal obesity, age was not statistically significant in multilevel analysis because most of our sample were between the ages of 22–24 years. Engaging in physical activity is protective against abdominal obesity in the general population and in AYLHIV as documented in other LMIC settings showing reduced odds of obesity and dyslipidaemia in those who engaged in at least moderate forms of physical

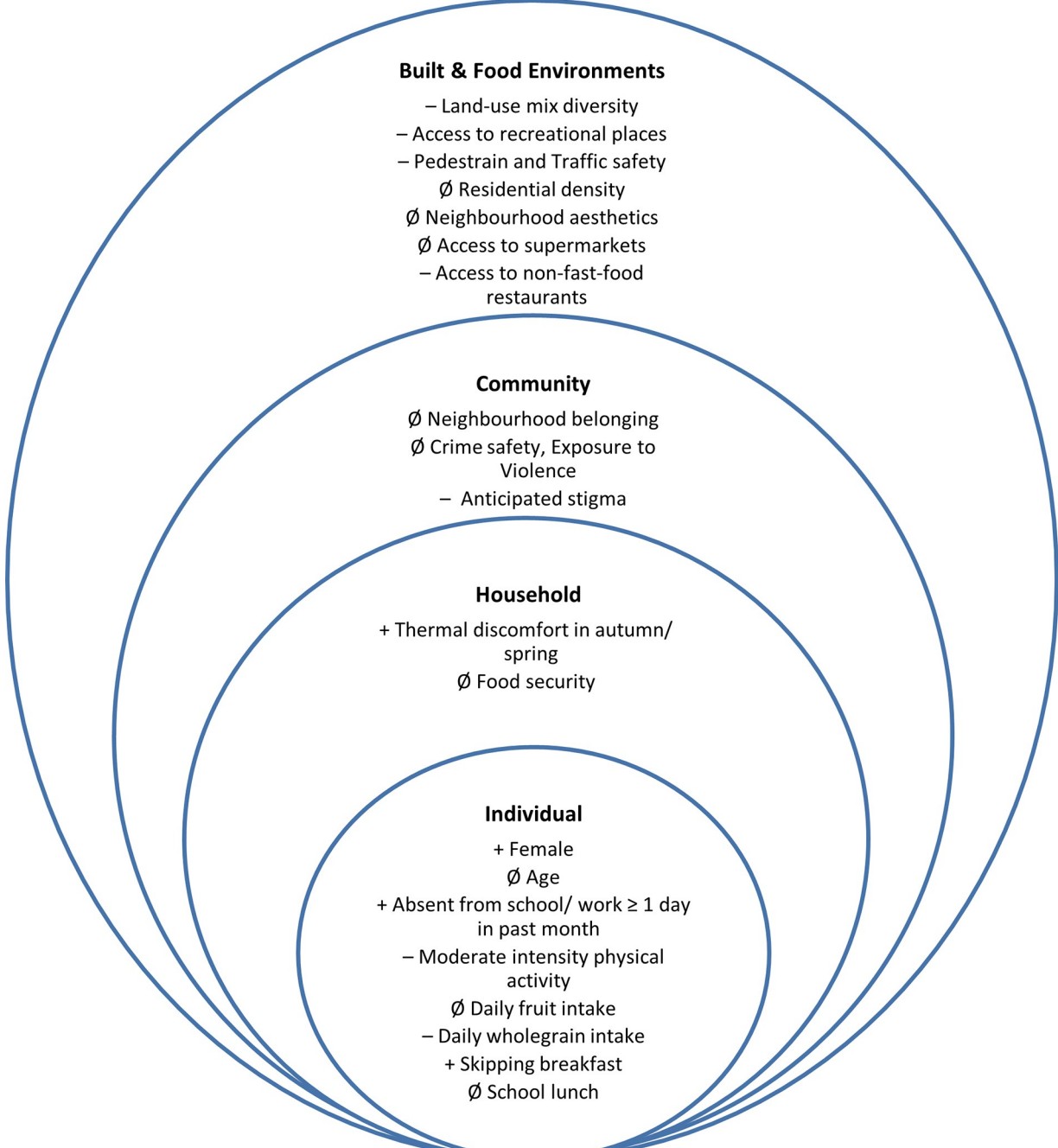

**Fig 1. Multilevel factors associated with abdominal obesity.** Covariates included in Fig 1 if statistically significant ($p < 0.10$) in bivariate regression. In multivariate regression: Ø no significant association; + significant positive association;–significant negative association.

activity [43]. Wholegrain intake is indicative of regularly eating breakfast which reduces the odds of obesity [41]. Skipping breakfast has become more prevalent among school-age children, adolescents and working adults [46] and was associated with higher odds of abdominal obesity in our study. Several studies have established an association between skipping breakfast and obesity [41]. A recent meta-analysis showed that skipping breakfast increased the risk of abdominal obesity by 31% [47].

**Table 4. Bivariate and multivariate associations between individual-level factors and abdominal obesity.**

| Explanatory variables OR [95% CI] | | Bivariate logistic regression | p-value | Multilevel model | p-value |
|---|---|---|---|---|---|
| Female | | 4.57 (1.21; 17.21) | **0.025***  | 3.02 (0.89; 10.23) | 0.076 |
| Age group | 15–17 years | 1.00 (ref) | | | |
| | 18–19 years | 4.125 (0.73; 23.43) | 0.110 | 4.48 (0.74; 27.26) | 0.104 |
| | 20–21 years | 4.65 (0.84; 25.66) | **0.077***  | 5.39 (0.87; 33.50) | 0.071 |
| | 22–24 years | 5.13 (0.96; 27.36) | **0.055***  | 3.87 (0.63; 23.58) | 0.143 |
| Ever pregnant or impregnated someone | | 3.77 (1.29; 11.01) | **0.015***  | 2.97 (0.79; 11.22) | 0.108 |
| Educational attainment: completed secondary school | | 0.94 (0.39; 2.26) | 0.886 | | |
| Neither in education employment nor training (NEET) | | 0.96 (0.36; 2.58) | 0.942 | | |
| Ever repeated a grade at school | | 1.64 (0.67; 4.02) | 0.276 | | |
| Days absent from school or work in past month | 0 days | 1.00 (ref) | | 1.00 (ref) | |
| | 1 or more days | 2.56 (1.06; 6.18) | **0.036***  | 3.06 (1.11; 8.40) | **0.030***  |
| Individual level multidimensional poverty (YMPI poor) | | 1.94 (0.80; 4.71) | 0.145 | 1.69 (0.63; 4.54) | 0.299 |
| Blood pressure | Normal BP | 1.00 (ref) | | 1.00 (ref) | |
| | Elevated BP | 2.04 (0.65; 6.34) | 0.219 | 2.37 (0.66; 8.51) | 0.185 |
| | Hypertension | 1.19 (0.18; 7.64) | 0.856 | 1.77 (0.21; 14.97) | 0.599 |
| Family history of diabetes | | 0.70 (0.26; 1.89) | 0.481 | 0.94 (0.30; 2.94) | 0.909 |
| **Physical activity per week** | | | | | |
| Any vigorous-intensity physical activity for ≥10 minutes | | 0.71 (0.28; 1.79) | 0.474 | | |
| Any moderate-intensity physical activity for ≥10 minutes | | 0.39 (0.13; 1.18) | **0.097***  | 0.24 (0.06; 0.92) | **0.038***  |
| Walking or cycling for ≥ 10 minutes | | 0.77 (0.29; 2.10) | 0.619 | | |
| Insufficient physical activity (< 600 MET-minutes/week) | | 0.73 (0.28; 1.88) | 0.509 | | |
| High physical activity (≥3000 MET minutes per week) | | 0.95 (0.35; 2.60) | 0.923 | | |
| Three or more hours sedentary per day | | 1.41 (0.58; 3.42) | 0.442 | 1.77 (0.67; 4.701) | 0.253 |
| **Current enrolment in school, college, university or working** | | 0.73 (0.29; 1.81) | 0.492 | | |
| Commuting time is less than 20 minutes | | 0.55 (0.17; 1.81) | 0.329 | | |
| Active Transport (Any walking or cycling in daily commute) | | 0.51 (0.16; 1.57) | 0.238 | | |
| Additional transport mode (more than one means) | | 1.36 (0.44; 4.20) | 0.589 | | |
| **Dietary intake and behaviour** | | | | | |
| Daily intake of fruits | | 0.35 (0.11; 1.08) | **0.067***  | 0.32 (0.074; 1.41) | 0.133 |
| Daily intake of vegetables | | 0.90 (0.36; 2.25) | 0.822 | | |
| Daily intake of wholegrain bread or cereal | | 0.30 (0.10; 0.88) | **0.028***  | 0.20 (0.054; 0.71) | **0.013***  |
| Daily intake of deep-fried foods | | 1.24 (0.41; 3.75) | 0.697 | | |
| Daily intake of fast foods | | 0.52 (0.14; 1.90) | 0.321 | | |
| Daily intake of sugar-sweetened beverages | | 1.00 (0.37; 2.70) | 1.000 | | |
| Daily intake of processed items with added sugar | | 1.24 (0.48; 3.24) | 0.657 | | |
| Ate meal prepared outside the home in past week | | 1.07 (0.40; 2.88) | 0.897 | | |
| Meals eaten outside home in past week | | 0.97 (0.80; 1.16) | 0.722 | | |
| Breakfast consumption | Non-skippers | 1.00 (ref) | | 1.00 (ref) | |
| | Semi-skippers | 6.35 (1.69; 23.88) | **0.006***  | 6.59 (1.51; 28.79) | **0.012***  |
| | Skippers | 3.63 (1.14; 11.56) | **0.029***  | 5.42 (1.32; 22.25) | **0.019***  |
| School serves school lunch (n = 38 in high school) | | 0.23 (0.057; 0.89) | **0.034***  | **0.23 (0.038; 1.408)** | **0.112** |
| **General Nutrition Knowledge Score** | | 0.99 (0.95; 1.05) | 0.962 | | |
| 1. Dietary recommendations (score/18) | | 0.96 (0.81; 1.12) | 0.587 | | |
| 2. Food Groups (score/36) | | 0.99 (0.88; 1.12) | 0.859 | | |
| 3. Healthy Food choices (score/13) | | 1.05 (0.86; 1.27) | 0.658 | | |
| 4. Diet, disease relationships (score/21) | | 1.04 (0.89; 1.22) | 0.594 | | |

*P-values less than 0.10 in bivariate logistic regression or less than 0.05 in multilevel logistic regression are shown in bold.

**Table 5. Bivariate and multivariate associations between household, community, built and food environment factors and abdominal obesity.**

| Explanatory variables OR [95% CI] | | Bivariate logistic regression | p-value | Multilevel model | p-value |
|---|---|---|---|---|---|
| **Family structure** | | | | | |
| Death of a parent | | 1.19 (0.50; 2.84) | 0.692 | | |
| Biological parent(s) as primary caregiver | | 1.32 (0.56; 3.10) | 0.527 | | |
| Number of people residing in the same house | | 0.94 (0.83; 1.07) | 0.341 | | |
| Household working-age adult employment | | 1.50 (0.44; 5.09) | 0.516 | | |
| **Household food security** | food secure | 1.00 (ref) | | | |
| | mild food insecurity | 1.25 (0.29; 5.35) | 0.764 | 1.77 (0.33; 9.46) | 0.504 |
| | moderate food insecurity | 0.63 (0.18; 2.16) | 0.457 | 0.62 (0.15; 2.47) | 0.496 |
| | severe food insecurity | 0.86 (0.30; 2.41) | 0.767 | 0.96 (0.30; 3.00) | 0.938 |
| **Housing characteristics** | | | | | |
| Formal dwelling | | 1.00 (ref) | | | |
| Informal shack /backyard dwelling | | 0.97 (0.39; 2.40) | 0.949 | | |
| Residential Stability | never moved | 1.00 (ref) | | | |
| | moved once | 0.87 (0.31; 2.44) | 0.792 | | |
| | moved two or more times | 2.13 (0.60; 7.62) | 0.243 | | |
| Dwelling ever damaged by negative event | | 1.19 (0.39; 3.66) | 0.765 | | |
| Thermal discomfort in summer | | 1.77 (0.72; 4.35) | 0.213 | | |
| Thermal discomfort in autumn or spring | | 3.01 (1.20; 7.60) | **0.019***  | 4.42 (1.43; 13.73) | **0.010*** |
| Thermal discomfort in winter | | 2.10 (0.82; 5.35) | 0.120 | | |
| Household assets deprivation (does not own more than two essential assets) | | 1.25 (0.47; 3.31) | 0.654 | | |
| **Access to amenities** | | | | | |
| Households not using electricity or gas for heating[β] | | 0.92 (0.36; 2.38) | 0.868 | | |
| Households without a flush toilet | | 1.07 (0.22; 5.09) | 0.934 | | |
| Households without piped water on site | | 0.75 (0.23; 2.47) | 0.639 | | |
| Household waste removed weekly by local authorities | | 1.25 (0.48; 3.27) | 0.649 | | |
| **Social networks**: Belongs to an extra-mural group | | 0.91 (0.38; 2.16) | 0.829 | | |
| High neighbourhood trust | | 0.64 (0.26; 1.56) | 0.325 | | |
| High neighbourhood reciprocity | | 0.85 (0.30; 2.43) | 0.767 | | |
| High neighbourhood friendliness | | 0.83 (0.23; 2.95) | 0.769 | | |
| High neighbourhood belonging | | 2.58 (0.84; 7.92) | **0.097*** | 2.68 (0.81; 8.89) | 0.107 |
| **Overall stigma score** | | 0.89 (0.68; 1.17) | 0.411 | | |
| Anticipated stigma | | 0.60 (0.36; 1.00) | **0.051*** | 0.58 (0.33; 1.00) | **0.052*** |
| Enacted stigma | | 0.93 (0.47; 1.85) | 0.843 | | |
| Internalised stigma | | 1.39 (0.73; 2.64) | 0.317 | | |
| **Community violence score** | | 1.13 (0.93; 1.37) | 0.212 | | |
| No or little violence (score 0–1) | | 1.00 (ref) | | 1.00 (ref) | |
| Moderate level of violence (score 2–3) | | 0.77 (0.18; 3.30) | 0.726 | 0.71 (0.13; 3.81) | 0.690 |
| High level of violence (score ≥ 4) | | 1.61 (0.47; 5.44) | 0.446 | 2.79 (0.59; 13.26) | 0.197 |
| Perceived high crime rate in neighbourhood | | 1.24 (0.44; 3.51) | 0.689 | | |
| Crime rate makes it unsafe to walk at night | | 1.55 (0.54; 4.41) | 0.415 | | |
| Worried to be outside alone around home area | | 1.05 (0.41; 2.66) | 0.924 | | |
| Worried to be outside with someone around home area | | 1.17 (0.44; 3.11) | 0.755 | | |
| Worried to be or walk around neighbourhood alone or with friends | | 1.10 (0.43; 2.79) | 0.846 | | |
| Worried about being in a local or nearby park | | 0.63 (0.24; 1.69) | 0.362 | | |
| **Built Environment: Neighbourhood Environment Walkability Scale** | | | | | |
| A. Land use mix-diversity | | 0.56 (0.31; 1.00) | **0.050*** | 0.52 (0.27; 0.97) | **0.039*** |

*(Continued)*

**Table 5.** (Continued)

| Explanatory variables OR [95% CI] | Bivariate logistic regression | p-value | Multilevel model | p-value |
|---|---|---|---|---|
| B. Access to recreational places | 0.41 (0.21; 0.77) | **0.006***  | 0.37 (0.18; 0.74) | **0.005*** |
| C. Residential density (types of homes) | 0.55 (0.28; 1.08) | **0.082*** | 0.51 (0.24; 1.09) | 0.081 |
| D. Land use mix-access (access to services) | 0.83 (0.39; 1.77) | 0.634 | | |
| E. Street connectivity | 1.37 (0.74; 2.51) | 0.314 | | |
| F. Walking or cycling facilities | 0.89 (0.51; 1.55) | 0.688 | | |
| G. Neighbourhood aesthetics | 0.57 (0.33; 0.99) | **0.044*** | 0.64 (0.34; 1.19) | 0.161 |
| H. Pedestrian and traffic safety | 0.26 (0.08; 0.89) | **0.031*** | 0.20 (0.05; 0.80) | **0.023*** |
| I. Crime safety | 1.14 (0.68; 1.93) | 0.615 | | |
| Food Environment: within walking distance[π] from home to selected destinations | | | | |
| Kiosk, corner store or small grocer | 0.99 (0.25; 3.85) | 0.985 | | |
| Supermarket within walking distance | 0.48 (0.19; 1.24) | 0.132 | 0.37 (0.12; 1.16) | 0.088 |
| Fruit or vegetable market within walking distance | 0.71 (0.16; 3.09) | 0.649 | | |
| Fast-food restaurant within walking distance | 0.67 (0.27; 1.70) | 0.401 | | |
| Non-fast-food restaurant within walking distance | 0.45 (0.18; 1.17) | 0.101 | 0.30 (0.10; 0.93) | **0.036*** |
| Coffee shop within walking distance | 0.57 (0.21; 1.54) | 0.269 | | |

*P-values less than 0.10 in bivariate logistic regression and less than 0.05 in multilevel logistic regression are shown in bold

[α] n = 46 in school or college

[β]Lighting fuel and cooking fuel deprived omitted because of collinearity

[π]Within walking distance defined as ≤ 20-minute walk from home.

The proliferation of fast-food outlets and consumption of SSBs in LMICs has increasingly been associated with childhood and adolescent obesity [48, 49]. However, we found no significant bivariate or multilevel associations between obesity and daily intake of unhealthy foods, SSBs, or eating food prepared outside the home suggesting that dietary intake, although a proximal factor that varied individually, may have more of a cumulative effect on obesity risk which warrants follow-up investigation over the life-course. Nutritional knowledge also did not differ by obesity status, which is important to note as an individual-level factor usually targeted by interventions. Absence from school or work was associated with increased odds of obesity, which potentially relates to access to nutrition through interventions like the National School Nutrition Programme (NSNP). The NSNP aims to reduce food insecurity and improve school-going children's health and nutritional status by providing nutritious meals and nutrition education [50]. Adolescents who benefit from school nutrition interventions have been documented to have beneficial outcomes, including changes in nutritional knowledge, dietary behaviours and physical activity [51]. Eating regular breakfast and school feeding initiatives are amenable approaches that can be encouraged with appropriate interventions.

Economists have written extensively about the association between obesity and socio-economic status (SES) in adults living in LMICs [52]. In settings where income inequality is high, the burden of obesity shifts to the most deprived in society [52]. In this study, those who were multidimensionally poor or experienced household food insecurity had similar odds of abdominal obesity compared to those not classified as poor or food insecure. In the South African context where undernutrition, inequality and obesity co-exist, the relationship between SES and obesity may not be linear. It may follow more of a social gradient where vulnerabilities and inequities are compounded over the life course in the most disadvantaged groups [53]. Interventions will need to address gaps and inequities using a life course approach starting from maternal and early childhood nutrition.

One unexpected finding was the association of abdominal obesity with thermal discomfort in autumn and spring and not in the more extreme temperature seasons of summer and winter. Research from higher-income settings has suggested that higher ambient temperatures are associated with increased odds of obesity [54, 55]. Thermal discomfort requires more detailed exploration as a potential contributor to obesity in diverse climatic LMIC settings such as Cape Town. Other housing characteristics such as access to amenities, and individual-level poverty measures that may be correlated with thermal discomfort were not significantly associated with abdominal obesity. This suggests that other unmeasured individual or household factors may play a mediating role in the relationship between thermal discomfort and obesity in the context of housing informality.

Children who live with grandparents or single mothers have been reported to have higher levels of overweight and obesity than those living with both parents, and children without siblings have a higher risk of obesity than children with siblings [56, 57]. However, these findings are predominantly from high-income settings, and this relationship might not be transferrable to LMIC settings. In our study, factors such as family structure, orphanhood status and primary caregiver relationship did not emerge as significantly related to obesity status. However, we did not measure parental/maternal obesity status, which has been found to be a better predictor of childhood obesity status and obesity in adulthood [58] as less than half of our participants lived with a biological parent. Further research is required to elucidate the influence of family structure on nutritional behaviours and obesity status in a non-nuclear family environment.

We found no association between crime safety, exposure to violence and obesity, which is contrary to previous studies that have suggested that high neighbourhood crime levels increase the risk of obesity in adolescents and adults [59, 60]. This is probably due to the homogeneity of our study population with participants reporting similar high levels of exposure to violence, which is endemic to informal and urban settings in Cape Town [61]. It is also likely that the effects of exposure to violence and crime might have a cumulative effect on health and lead to future obesity in adulthood, as demonstrated in a cohort of African-American youth for whom fear of neighbourhood violence during adolescence was predictive of obesity almost a decade later [62].

Measures of social capital did not have significant associations with obesity in multilevel analysis. However, higher perceived neighbourhood belonging showed a tendency toward increased odds of abdominal obesity. Although the mechanism by which social capital affects health and well-being is not clearly elucidated in LMIC contexts [63], one mechanism could be via peer influence. Young people may buy into and emulate the eating behaviours of neighbourhood social contacts, which in urbanised settings, often entails consuming more processed, low-quality foods and becoming part of new social networks where being overweight is more normative [7]. On the other hand, in the context of HIV, stigma is a critical form of negative social capital that plays out in the form of social exclusion of those living with HIV [64]. Our findings indicate that those who are not obese anticipate experiencing more stigma from the community which speaks to perceptions of body image in the South African context, where people perceive being overweight as desirable given the perceived association between being underweight and being HIV positive and also given cultural connotations of affluence and beauty [65, 66].

Several subscales of perceived walkability in the neighbourhood environment were significantly associated with reduced odds of abdominal obesity, including land-use mix diversity, access to recreational places, pedestrian and traffic safety, and access to non-fast-food restaurants. In previous studies conducted in the United States of America, neighbourhood environment walkability, particularly higher residential density, land use mix, street connectivity, and

aesthetics were associated with physical activity and lower obesity prevalence [67]. Being able to walk easily from home to commercial areas in neighbourhoods with mixed land use is associated with reduced body weight and increased walking and physical activity behaviours [68]. Our findings corroborate these findings in a lower-income setting and highlight the importance of a mix of residential, commercial, industrial, and open spaces in urban design to promote non-motorised forms of transport. Urban space diversity is crucial in the urban planning of a city like Cape Town with its history of displacement and continued spatial segregation [69, 70]. Access to recreational places also emerged as significantly associated with lower odds of abdominal obesity, similar to studies that document that access to recreational facilities is correlated with adolescent physical activity and weight status [71]. Proximity to parks and other recreational spaces may increase physical activity, improve HIV-related health outcomes, and reduce depression in PLHIV [72, 73]. However, this relationship may be mediated by SES and neighbourhood safety in urban, low-income settings [74]. Therefore, local governments need to consider developing parks and other open recreational spaces in urban areas with safety measures in place as a means to promote physical activity in young people [75].

Previous multilevel studies suggest that the availability of food stores is significantly related to individual-level obesity [68]. Access to supermarkets that sell healthy foods has been linked to improved dietary choices as people with access to supermarkets consume more fruits and vegetables than those who rely on convenience stores and corner shops [76, 77]. Except for non-fast-food restaurants, access to supermarkets, fruit and vegetable markets, and fast-food restaurants were not significantly related to abdominal obesity in our analysis. Our findings illustrate similar food environments across the communities studied. These areas typically have informal fruit and vegetable traders and convenience shops as part of their food landscape and very few supermarkets [78]. The emergence of non-fast food restaurants related to lower odds of abdominal obesity is an interesting finding in this context compared to other settings where proximity to non-fast food restaurants had no discernible effect on obesity or weight gain [79]. It is crucial to elucidate how young people navigate their food environments in this setting, especially in light of homogenous exposures to food advertising, availability and accessibility of foods. The South African government implemented mandatory legislation for salt reduction in processed foods [80] and a tax on SSB [81]. However, more measures are needed at a community and micro level to promote healthier food environments and encourage regular physical activity.

Few, if any studies have examined multilevel determinants of obesity in AYLHIV. Previous studies in PLHIV in LMICs have explored individual, particularly treatment-related factors related to obesity. The inclusion of environmental determinants in this study expands the understanding of obesity risk in AYLHIV beyond individual and healthcare factors. Furthermore, abdominal obesity measured using the WHtR reflects NCD risk more accurately than BMI in this population of AYLHIV. Our study had several limitations. Firstly, due to the limited sample size, we could not control for all possible confounders and experienced convergence and precision issues when this was attempted. Preliminary sample size calculations achieved a power of 80% to detect a prevalence of obesity of 5.5%. However, we may have missed smaller associations due to low statistical power. However, by using multilevel analysis, we accounted for the hierarchical structure of the data and neighbourhood-level differences. Second, our variables were based on self-report, which might be prone to reporting bias. Moreover, the measurements of thermal comfort, social capital and the food environment have not been previously validated in this population and may have reduced reliability in this setting. Lastly, we failed to pick up geospatial coordinates from the addresses provided by participants and used sub-districts as the grouping variable for neighbourhoods instead. Nonetheless, this

study provides a framework for more detailed future studies using multilevel methods in similar settings.

## Conclusion

We report that abdominal obesity is highly prevalent in AYLHIV in peri-urban Cape Town. This study adds to the limited body of literature addressing multilevel determinants of obesity in a population more vulnerable to NCDs. Our findings highlight factors across multiple levels from individual to neighbourhood environments, that affect obesity risk in AYLHIV. Obesity prevention efforts that target adolescents have the greatest potential to avert obesity into adulthood due to the critical nature of adolescence as a development period. Obesity continues to increase and is a major health issue in LMICs. We recommend that further research be conducted in AYLHIV in similar settings to generate contextually relevant evidence to effectively turn the tide of the obesity epidemic in rapidly urbanising LMIC cities. Important areas to explore include the role of a non-nuclear family structure on obesity risk, thermal discomfort and housing informality, social norms and community perceptions, food availability and urban design. The diverse range of interventions required highlights the importance of intersectoral action, engaging diverse sectors and actors to reduce obesity risk in this priority population group.

## Supporting information

**S1 Table. Definition and measurement of derived variables.**
(DOCX)

**S1 Fig. Enrolment by facility and sub-district.**
(TIF)

## Acknowledgments

The authors appreciate the assistance from the City of Cape Town and Western Cape Departments of Health who made access to the facilities and data possible. We would like to thank all the participants for their time and willingness to participate in this research.

## Author Contributions

**Conceptualization:** Monika Kamkuemah, Tolu Oni, Keren Middelkoop.

**Data curation:** Monika Kamkuemah, Blessings Gausi.

**Formal analysis:** Monika Kamkuemah.

**Funding acquisition:** Tolu Oni, Keren Middelkoop.

**Investigation:** Monika Kamkuemah, Blessings Gausi, Tolu Oni, Keren Middelkoop.

**Methodology:** Monika Kamkuemah, Tolu Oni, Keren Middelkoop.

**Project administration:** Monika Kamkuemah, Blessings Gausi, Tolu Oni.

**Resources:** Blessings Gausi, Tolu Oni, Keren Middelkoop.

**Supervision:** Tolu Oni, Keren Middelkoop.

**Writing – original draft:** Monika Kamkuemah, Tolu Oni, Keren Middelkoop.

**Writing – review & editing:** Monika Kamkuemah, Tolu Oni, Keren Middelkoop.

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
