## [Decision Letter · Decision Letter 0]

2 Sep 2022

PONE-D-22-08097Multilevel Correlates of Abdominal Obesity in Adolescents and Youth Living with HIV in peri-urban Cape Town, South Africa.PLOS ONE

Dear Dr. Kamkuemah,

Thank you for submitting your manuscript to PLOS ONE. After careful consideration, we feel that it has merit but does not fully meet PLOS ONE’s publication criteria as it currently stands. Therefore, we invite you to submit a revised version of the manuscript that addresses the points raised during the review process. We sincerely apologize for the delay in issuing this editorial decision. Unfortunately, we experienced difficulties in finding available reviewers and Academic Editors to assess your manuscript.

The manuscript has now been evaluated by one reviewer, and their comments are available below. The reviewer provided very positive feedback about your work and did not identify substantial changes to make. They have only suggested that you make minor changes to the statistical analysis section. Could you please revise the manuscript to carefully address the point raised?

Please note that since we have only been able to secure a single reviewer to assess your manuscript, we are issuing a decision at this point to prevent further delays. Please be aware that the editor who handles your revised manuscript might find it necessary to invite additional reviewers to assess this work once the revised manuscript is submitted. However, we will aim to proceed on the basis of this single review if possible. 

We look forward to receiving your revised manuscript.

Kind regards,

Dario Ummarino, PhD

Senior Editor

PLOS ONE

Reviewers' comments:

Reviewer's Responses to Questions

**Comments to the Author**

1. Is the manuscript technically sound, and do the data support the conclusions?

Reviewer #1: Yes

2. Has the statistical analysis been performed appropriately and rigorously? 

Reviewer #1: Yes

3. Have the authors made all data underlying the findings in their manuscript fully available?

Reviewer #1: Yes

4. Is the manuscript presented in an intelligible fashion and written in standard English?

Reviewer #1: Yes

5. Review Comments to the Author

Reviewer #1: The paper is well written.

The authors conducted a cross sectional study to investigate the prevalence of abdominal obesity and associated multilevel factors in AYLHIV in peri-urban Cape Town, South Africa. The methods are clear, results presented well, the discussion is sound, and there is coherence up to the conclusion.

The authors can just add the test they have used to check the distribution of variables under statistical analysis, which informed use of both non-parametric and parametric tests.

Excellent!

6. PLOS authors have the option to publish the peer review history of their article (what does this mean?). If published, this will include your full peer review and any attached files.

Reviewer #1: No

---

## [Author Response · Author response to Decision Letter 0]

21 Oct 2022

Response to Additional Journal Requirements

Comment 1. Please ensure that your manuscript meets PLOS ONE's style requirements, including those for file naming. The PLOS ONE style templates can be found at

 and

Response: The manuscript has now been formatted to meet PLOS ONE's style requirements. 

Comment 2. We note that you have included the phrase “data not shown” in your manuscript. Unfortunately, this does not meet our data sharing requirements. PLOS does not permit references to inaccessible data. We require that authors provide all relevant data within the paper, Supporting Information files, or in an acceptable, public repository. Please add a citation to support this phrase or upload the data that corresponds with these findings to a stable repository (such as Figshare or Dryad) and provide and URLs, DOIs, or accession numbers that may be used to access these data. Or, if the data are not a core part of the research being presented in your study, we ask that you remove the phrase that refers to these data.

Response: Thank you for pointing this out. All data are accessible on Figshare at this URL: https://figshare.com/articles/dataset/HIV_NCD_Abdominal_Obesity_Analysis_WHtR_dta/19204929

We have removed the phrase “data not shown” from the manuscript. 

Comment 3. Please review your reference list to ensure that it is complete and correct. If you have cited papers that have been retracted, please include the rationale for doing so in the manuscript text, or remove these references and replace them with relevant current references. Any changes to the reference list should be mentioned in the rebuttal letter that accompanies your revised manuscript. If you need to cite a retracted article, indicate the article’s retracted status in the References list and also include a citation and full reference for the retraction notice.

Response: We have reviewed and updated the reference list. It does not contain any retracted articles. 

Response to Reviewer 1

5. Review Comments to the Author

Reviewer #1: The paper is well written.

The authors conducted a cross sectional study to investigate the prevalence of abdominal obesity and associated multilevel factors in AYLHIV in peri-urban Cape Town, South Africa. The methods are clear, results presented well, the discussion is sound, and there is coherence up to the conclusion.

The authors can just add the test they have used to check the distribution of variables under statistical analysis, which informed use of both non-parametric and parametric tests.

Excellent!

Response: Thank you very much for this positive review. In response to the statistical tests used to check the distribution of variables, we have added the following to the methods section: “We used visual displays, histograms, box plots and the Shapiro–Wilk test to test for normality of the continuous variables.”

---

## [Decision Letter · Decision Letter 1]

23 Nov 2022

PONE-D-22-08097R1Multilevel correlates of abdominal obesity in adolescents and youth living with HIV in peri-urban Cape Town, South Africa.PLOS ONE

Dear Dr. Kamkuemah,

Thank you for submitting your manuscript to PLOS ONE. After careful consideration, we feel that there are only a couple minor revisions necessary before acceptance for publication, namely those recommended by reviewer 2. 

We look forward to receiving your revised manuscript.

Kind regards,

Blake Byron Walker, Ph.D.

Academic Editor

PLOS ONE

Journal Requirements:

Reviewers' comments:

Reviewer's Responses to Questions

**Comments to the Author**

1. If the authors have adequately addressed your comments raised in a previous round of review and you feel that this manuscript is now acceptable for publication, you may indicate that here to bypass the “Comments to the Author” section, enter your conflict of interest statement in the “Confidential to Editor” section, and submit your "Accept" recommendation.

Reviewer #1: All comments have been addressed

Reviewer #2: (No Response)

2. Is the manuscript technically sound, and do the data support the conclusions?

Reviewer #1: Yes

Reviewer #2: Yes

3. Has the statistical analysis been performed appropriately and rigorously? 

Reviewer #1: Yes

Reviewer #2: Yes

4. Have the authors made all data underlying the findings in their manuscript fully available?

Reviewer #1: Yes

Reviewer #2: Yes

5. Is the manuscript presented in an intelligible fashion and written in standard English?

Reviewer #1: Yes

Reviewer #2: Yes

6. Review Comments to the Author

Reviewer #1: All corrections have been addressed and well implemented. I commend the authors for this kind of a research work.

No dual publication discovered.

Ethics has been granted for this study.

Authors have adhered to publication ethics.

Reviewer #2: Methods

Were there any differences in findings from the study sites, that is, the clinics where participants were recruited from? Of the 87 participants, what were the distributions in numbers from each clinic (study site)? Were all interviews conducted in English?

Statistical analysis

Line 235: Variables found to be associated with abdominal obesity in bivariate analysis (p < 0.10) and variables identified a priori in the literature were included in the multilevel models. Could you please indicate those variables that were identified a priori in the literature. This will inform the reader as to why some variables in table 4 and table 5 are included in the multilevel model when the bivariate p value is >0.10.

Line 240: We also checked the intra-class correlation coefficient (ICC) to analyse the variability within and between sub-districts. Was there variability, this has not been discussed.

Table 1: The following variables do not add up to the total (n):

Days absent from school

Dwelling type

Deceased parents

Table 2: Abbreviations SSBs and MET to be written in full in the caption so that the table is self-explanatory.

7. PLOS authors have the option to publish the peer review history of their article (what does this mean?). If published, this will include your full peer review and any attached files.

Reviewer #1: **Yes: **Prof Perpetua Modjadji

Reviewer #2: No

---

## [Author Response · Author response to Decision Letter 1]

7 Jan 2023

Reviewer's Responses to Questions

Comments to the Author

1. If the authors have adequately addressed your comments raised in a previous round of review and you feel that this manuscript is now acceptable for publication, you may indicate that here to bypass the “Comments to the Author” section, enter your conflict of interest statement in the “Confidential to Editor” section, and submit your "Accept" recommendation.

Reviewer #1: All comments have been addressed

Reviewer #2: (No Response)

2. Is the manuscript technically sound, and do the data support the conclusions?

Reviewer #1: Yes

Reviewer #2: Yes

3. Has the statistical analysis been performed appropriately and rigorously?

Reviewer #1: Yes

Reviewer #2: Yes

4. Have the authors made all data underlying the findings in their manuscript fully available?

The PLOS Data policy requires authors to make all data underlying the findings described in their manuscript fully available without restriction, with rare exception (please refer to the Data Availability Statement in the manuscript PDF file). The data should be provided as part of the manuscript or its supporting information or deposited to a public repository. For example, in addition to summary statistics, the data points behind means, medians and variance measures should be available. If there are restrictions on publicly sharing data—e.g. participant privacy or use of data from a third party—those must be specified.

Reviewer #1: Yes

Reviewer #2: Yes

5. Is the manuscript presented in an intelligible fashion and written in standard English?

Reviewer #1: Yes

Reviewer #2: Yes

6. Review Comments to the Author

Reviewer #1: All corrections have been addressed and well implemented. I commend the authors for this kind of a research work.

No dual publication discovered.

Ethics has been granted for this study.

Authors have adhered to publication ethics.

Response: Thank you very much for this positive feedback and for agreeing to review the revised manuscript. 

Response to Reviewer 2

Comment 1: Methods

Were there any differences in findings from the study sites, that is, the clinics where participants were recruited from? Of the 87 participants, what were the distributions in numbers from each clinic (study site)? Were all interviews conducted in English?

Response: We did not have sufficient statistical power to detect differences across the six clinics (< 30 participants were recruited in each clinic). Hence, we used the sub-districts as the cluster variable. The distribution of participants recruited from each site is included as a supplementary figure (S1 Figure). All interviews were conducted in English with translations into the local languages provided where necessary. 

Comment 2: Statistical analysis

Line 235: Variables found to be associated with abdominal obesity in bivariate analysis (p < 0.10) and variables identified a priori in the literature were included in the multilevel models. Could you please indicate those variables that were identified a priori in the literature. This will inform the reader as to why some variables in table 4 and table 5 are included in the multilevel model when the bivariate p value is >0.10.

Response: We have added the following to clarify which variables we are referring to: “The following variables identified a priori to be associated with obesity were included in the multilevel models: multidimensional poverty (YMPI), blood pressure, family history of diabetes, sedentary behaviour, physical activity, dietary intake, household food security, and level of community violence.”

Comment 3: Line 240: We also checked the intra-class correlation coefficient (ICC) to analyse the variability within and between sub-districts. Was there variability, this has not been discussed.

Response: The ICC for the fitted models was generally low, (<0.2) indicating that there was variability between and within sub-districts. We have added this sentence to the results section. “The ICC ranged from 0.01 to 0.2, indicating some level of variability between and within sub-districts.” 

Comment 4: Table 1: The following variables do not add up to the total (n):

Days absent from school

Dwelling type

Deceased parents

Response: There were some missing values for these variables. This has been indicated in the first columns in the revised the table as follows: “Days absent from school: n= 86, Dwelling type: n= 86 and Deceased parents: n= 50. 

Comment 5: Table 2: Abbreviations SSBs and MET to be written in full in the caption so that the table is self-explanatory.

Response: Table 2 now includes the full descriptions for the abbreviations used in the table in the footnote placed below the table as per journal requirements. PA= physical activity, MET= metabolic equivalent of task, SSB= sugar sweetened beverages. 

7. PLOS authors have the option to publish the peer review history of their article (what does this mean?). If published, this will include your full peer review and any attached files.

Do you want your identity to be public for this peer review? For information about this choice, including consent withdrawal, please see our Privacy Policy.

Reviewer #1: Yes: Prof Perpetua Modjadji

Reviewer #2: No

---

## [Editor Report · Decision Letter 2]

11 Jan 2023

Multilevel correlates of abdominal obesity in adolescents and youth living with HIV in peri-urban Cape Town, South Africa.

PONE-D-22-08097R2

Dear Dr. Kamkuemah,

We’re pleased to inform you that your manuscript has been judged scientifically suitable for publication and will be formally accepted for publication once it meets all outstanding technical requirements.

Kind regards,

Blake Byron Walker, Ph.D.

Academic Editor

PLOS ONE
---

## [Editor Report · Acceptance letter]

16 Jan 2023

PONE-D-22-08097R2 

Multilevel correlates of abdominal obesity in adolescents and youth living with HIV in peri-urban Cape Town, South Africa 

Dear Dr. Kamkuemah:

I'm pleased to inform you that your manuscript has been deemed suitable for publication in PLOS ONE. Congratulations! Your manuscript is now with our production department. 

Kind regards, 

on behalf of

Prof. Dr. Blake Byron Walker 

Academic Editor

PLOS ONE